# Plant Virus Nanoparticles Combat Cancer

**DOI:** 10.3390/vaccines11081278

**Published:** 2023-07-25

**Authors:** Mehdi Shahgolzari, Srividhya Venkataraman, Anne Osano, Paul Achile Akpa, Kathleen Hefferon

**Affiliations:** 1Drug Applied Research Center, Tabriz University of Medical Sciences, Tabriz 5166616471, Iran; 2Department of Cell & Systems Biology, University of Toronto, Toronto, ON M5S 3B2, Canada; 3Department of Natural Sciences, Bowie State University, Bowie, MD 20715, USA; 4Department of Pharmaceutics, Faculty of Pharmaceutical Sciences, University of Nigeria, Nsukka 410001, Enugu State, Nigeria; 5Department of Microbiology, Cornell University, Ithaca, NY 14850, USA

**Keywords:** nanoparticles, plant virus-like particles, vaccines, delivery

## Abstract

Plant virus nanoparticles (PVNPs) have garnered considerable interest as a promising nanotechnology approach to combat cancer. Owing to their biocompatibility, stability, and adjustable surface functionality, PVNPs hold tremendous potential for both therapeutic and imaging applications. The versatility of PVNPs is evident from their ability to be tailored to transport a range of therapeutic agents, including chemotherapy drugs, siRNA, and immunomodulators, thereby facilitating targeted delivery to the tumor microenvironment (TME). Furthermore, PVNPs may be customized with targeting ligands to selectively bind to cancer cell receptors, reducing off-target effects. Additionally, PVNPs possess immunogenic properties and can be engineered to exhibit tumor-associated antigens, thereby stimulating anti-tumor immune responses. In conclusion, the potential of PVNPs as a versatile platform for fighting cancer is immense, and further research is required to fully explore their potential and translate them into clinical applications.

## 1. Introduction

Nanomedicine, a burgeoning area of multidisciplinary research, has exhibited enormous potential for transformation into a highly innovative development [1]. Despite the existence of numerous products in clinical trials and on pharmacy shelves worldwide, usage remains limited due to the rather prohibitive price tags of these innovative products [2]. It is noteworthy to state that although significant advancements have been suggested in terms of the anticipated efficacy of nanomedicines, opinions tend to differ regarding the stage of critical cost–benefit analysis for the accessibility of nanomedicines in the treatment of cancer and other maladies.

Nanomedicines encompass a broad range of nanomaterials, showcasing a particle size that spans from 1 nm to more than 400 nm, and represent a highly diversified array of materials. These materials may consist entirely of metal, as exemplified by gold and silver nanoparticles [3], or may be composed of a blend of liquids or a ternary system that is constituted by a combination of various compatible materials, resulting in a multifunctional entity, which often exhibits stimuli-responsive attributes that enable it to react to minute changes in factors such as pH and temperature variations [4]. Furthermore, nanoparticles can be fabricated using simple polymeric materials, including cellulose and chitosan. Recently, proteinaceous nanoparticles made from bacteriophage, plant, and mammalian viruses due to their capacity for immunomodulation, also offer special advantages [5].

Plant VNPs have been examined as an exclusive category of nanocarriers for biomedical applications, as documented in previous research [6]. In addition to their facile production and continuous quality control maintenance, plant virus VNPs provide a rational substitute for synthetic nanoparticles due to their economical nature, non-toxicity, and biodegradability. Moreover, plant VNPs have been enhanced to improve their performance with regard to stimuli-responsivity, as evidenced by recent developments [7].

Plant VNPs exhibit either a rod shape (tobacco mosaic virus (TMV) and potato virus X (PVX)) or an icosahedral shape (cowpea mosaic virus (CPMV)). These differing shapes elicit distinct nanoparticle responses in vivo. Tobacco mosaic virus is capable of assembling into VLPs without necessitating drug payload conjugation on the nanoparticle surface or within its inner channel, albeit to a limited degree. In contrast, potato virus X requires its RNA genome for self-assembly and can only carry a payload on its outer surface. Cowpea mosaic virus, on the other hand, can self-assemble into empty virus-like particles in the absence of its RNA genome and can therefore contain a payload both internally and externally within its protein shell.

Despite being non-infectious in humans, the multivalent, repetitive coat protein assemblies of the VNPs and VLPs function as pathogen-associated molecular patterns (PAMPs), serving as danger signals to the immune system [8]. VNPs administered through intramuscular and subcutaneous routes drain effectively into lymph nodes and stimulate immune cells following recognition by the cellular pattern recognition receptors. VNP-derived vaccines also enable antigen cross-presentation which is vital for the major histocompatibility complex class I (MHC-I)-mediated presentation of extracellular antigens to elicit a potent cytotoxic T-cell response.

For instance, TMV, PVX, and papaya mosaic virus VNPs are capable of eliciting robust cellular immune responses against fused antigenic epitopes [9]. The potent immunostimulatory characteristics of the VNPs have been shown for a CPMV vaccine that was remarkably efficient in reducing tumors such as glioma, melanoma, ovarian, colon, and breast cancers in animal models [10,11,12,13,14]. In these studies, the CPMV VNPs were directly administered into the tumor to activate the innate immune cells present within the tumor microenvironment which prime killing of the tumor cells as well as antigen processing which leads to the development of systemic antitumor immunity. Also, as against oncolytic viral tumor therapy, the occurrence of pre-existing immunity does not compromise the efficiency of the immune response elicited by the plant VNPs [15]. Rather, antibody recognition augments the opsonization of CPMV, resulting in the improvement of viral recognition by innate immune cells.

Hence, VNPs function as exemplary epitope delivery vehicles for antigens and adjuvants of great applicability in vaccine and immunotherapeutic formulations [16]. The present preclinical advancement pipeline for these VNP-based vaccines includes autoimmune diseases, substance abuse, and cardiovascular and infectious diseases [15]. Furthermore, plant VNP platforms hold great promise as epidemic or pandemic vaccines [17] due to their increased thermal stability and therefore would not be subject to the requirements of cold chain and can be easily produced through molecular farming.

In this review, we present a multitude of examples to analyze the manner in which the structural composition of plant viruses contributes to their efficacious utilization in the realm of cancer diagnostics and therapy. Furthermore, we delve into the origins of the implementation of plant virus architecture as nanoparticles in medical applications, along with the prospects of their propitious employment as pioneering cancer immunotherapies.

## 2. Tobacco Mosaic Virus (TMV)

For over a century, the history of the tobacco mosaic virus (TMV) has been significant and noteworthy. It all started when Beijerinck identified the mosaic disease of tobacco as a fluid that is capable of spreading, which was later referred to as a “virus” in modern language [18]. TMV got its name from one of the first plants where it was discovered in the 1800s. However, it can infect over 350 diverse plant species. TMV usually infects tobacco, solanaceous crops such as pepper and tomato, vine vegetables like cucumber, melon, and squash, and various ornamental plants like begonia, coleus, geranium, impatiens, million bells, and petunia [19]. TMV is the first identified virus and has significantly contributed to answering essential queries about the general nature of viruses [20].

TMV represents the Tobamovirus genus and is part of the alphavirus-like supergroup. The virions of TMV are rod-shaped, measuring 300 × 18 nm, and have a hollow center that contains 95% capsid protein (CP) and 5% RNA. The capsid protein (CP) subunits interact with 3 nts in a helical pattern around the RNA. The virions remain stable for many decades and their ability to infect sap remains unchanged even after being heated to 90 °C [21].

TMV’s application as a prototype system has been driving the field of virology research even in modern times. TMV was the first virus to go through chemical purification [22], detection in an analytical ultracentrifuge and electrophoresis device [23], as well as visualization under an electron microscope [24]. Additionally, TMV RNA was essential in the initial conclusive experiments that established the genetic information carried by nucleic acids and their sufficiency without any other component for viral infectivity [25]. The TMV coat protein (CP) was the initial virus protein to undergo sequencing [26], and the structure of TMV particles was one of the first to be revealed in atomic detail [27].

The genome of TMV, consisting of a single positive-sense RNA molecule, spans 6395 nucleotides (nts) and encompasses four open reading frames (ORF) [28]. The 5′-terminus of the TMV genomic RNA is capped with 7-methyl guanosine [29]. The 5′-proximal ORFs, encoding the overlapping 126 and 183 kDa replication proteins, initiate at nt 69 and terminate with amber and ochre stop codons at nt 3417–3419 and nt 4917–4919, respectively. Both proteins are directly translated from the genomic RNA, which ends with a terminal tRNA-like structure that specifically aminoacylates with histidine. The 5′-NTR serves as a strong translational enhancer [30]. Both the 5′- and 3′-NTRs contain cis-acting elements essential for replication [31].

TMV is presently being employed extensively as a tool for biological research and biotechnology applications [32]. Various host factors that interact with diverse TMV gene products have been identified, in addition to the cloning and characterization of one of the classic plant genes conferring resistance. TMV is currently being utilized as a medium to deliver and express foreign sequences in plants and as a model system to scrutinize virus–host interactions. Further innovations are being devised to explicate the other facets of TMV [33].

## 3. Potato Virus X

PVX is a plant virus that belongs to the Potexvirus genus of the Alphaflexiviridae family. It is regarded as the type member and is one of the oldest known viruses to infect potato plants, with scientific publications dating back to [32]. This virus causes a mild mosaic disease in solanaceous plants such as tomatoes, tobacco, and potato. It is important to note that many viruses within the Potexvirus genus, which comprises 48 species, can cause severe diseases in their hosts [34]. PVX is prevalent worldwide and is usually transmitted through pollen or true seeds, contaminated farming equipment, or contact between healthy and infected foliage and roots [34]. It is worth noting that there is no evidence to suggest that PVX has an invertebrate vector.

PVX is a type of plant virus whose virions are made up of flexuous filaments that measure about 460–480 nm in length, 13 nm in diameter, and with a helical pitch of 3.4 nm. The genome of PVX is around 6.4 kb long and has both a 5′ cap and a 3′ poly(A) terminus. It consists of five open reading frames, with four considered essential for cell-to-cell and systemic movement, including one that encodes the viral replicase. Infected cells tend to have X-bodies, or cellular inclusions located near the nucleus, commonly observed in their vicinity [29].

PVX is recognized as one of the top 10 plant viruses in molecular plant pathology due to its significant contributions as a model for studying plant–virus interactions. It has played a crucial role in advancing plant biology, including the development of antiviral gene silencing mechanisms, creating viral vectors for foreign gene expression, using green fluorescent protein (GFP) to track plant virus infection, discovering the function of viral movement proteins in transport through plasmodesmata, and cloning the Rx gene in potato through map-based cloning. The Rx gene is considered the first antiviral R gene isolated from potato [35,36,37].

PVX has greatly contributed to the initial characterization of the homology-dependent post-transcriptional gene silencing (PTGS) pathway, which is responsible for destroying viral RNAs. The PTGS mechanism targets highly structured regions of viral genomes or replication intermediates that can be cleaved by plant-specific Dicer enzymes, resulting in the formation of siRNAs (small interfering RNAs). These studies were crucial and involved using PVX as an expression vector carrying cDNAs that encode foreign sequences such as green fluorescent protein (GFP), enabling the monitoring of siRNA production and mRNA turnover [38,39].

Improved PVX-based vectors are considered suitable for expressing any desired gene in PVX host plants. These vectors are favorable for protein overexpression, investigation of protein functions in plants, or utilization in virus-induced gene silencing mechanisms [40].

## 4. Cowpea Mosaic Virus (CPMV)

The CPMV belongs to the Comovirus genus and is the representative member of this group. The Comovirus genus features a total of 13 other members that are part of the larger Comoviridae family [41]. The family Comoviridae encompasses three different types of viruses: Comovirus, Fabavirus, and Nepovirus, all of which are roughly 30 nanometers (nm) in diameter and possess an isometric shape. Their genetic material consists of two single-stranded RNAs that are each located in a separate yet identical virus particle. Additionally, some virus particles may not contain any RNA at all. Furthermore, the protein shell for each virus particle comprises one, two, or three variations of protein subunits [42].

CPMV is responsible for causing one of the most commonly reported virus diseases in cowpea plants, resulting in chlorotic spots with indistinct borders on primary leaves and a yellow or light green mosaic pattern on trifoliate leaves, especially in younger ones. Its host range is limited mostly to plants belonging to Leguminosae, and it was first discovered in Nigeria in 1959 in an infected cowpea plant. Since then, it has been found in several other countries, including Kenya, Japan, Tanzania, Surinam, and Cuba. Although its natural host is cowpea, it can also infect other legumes, and *Nicotiana benthamiana* has proved to be an important experimental host. The virus is transmitted by various beetles that have biting mouthparts, and it has been reported in Africa, the Philippines, and Iran, but not in North and South America [42].

CPMV has a bipartite genome consisting of two separate RNA strands, RNA-1 and RNA-2, which are encapsulated separately. The virus possesses an icosahedral capsid structure with a diameter of approximately 30 nm, composed of 60 copies each of small (S) and large (L) coat proteins. These coat proteins are generated from a VP60 precursor polyprotein encoded by RNA-2, through action of the 24K viral proteinase found in RNA-1. Both RNA segments are required for successful viral infection and capsid assembly within a host plant cell [43]. Purified virus preparations contain three distinct components: empty protein shells lacking RNA (T), as well as two nucleoprotein components (M and B) containing 24% and 34% RNA, respectively. The heat-stable icosahedral particles have a diameter of 28 nm and are made up of 60 copies of the two coat proteins [44].

CPMV is a plant virus that has gained significant attention in research. Initially, studies were focused on understanding the virus’s genetic makeup and structure. However, current research has shifted towards exploring the potential of CPMV for biotechnology purposes. One such application involves using CPMV as a platform to present antigenic epitopes from various sources, including infectious agents and tumors, to develop vaccines [45]. This research has led to the creation of replicating virus vectors and a highly efficient protein expression system that does not rely on viral replication. Additionally, CPMV has been developed as an expression and presentation system for peptides derived from vaccine targets [44].

## 5. Multifunctional Plant Virus Nanoparticles (PVNPs) in Cancer

Plant viruses consist of a genome encapsulated in a capsid formed from multiple copies of coat proteins (CPs) [46,47]. CPs can self-assemble into closed-caged structures without a genome as virus-like particles (VLP) [48]. Plant viruses (virions) and plant VLPs are considered plant virus nanoparticles (PVNPs) within various sizes (10–500 nm) and shapes (icosahedrons, rods, filaments) [49,50]. They are uniform, monodispersed, biocompatible, biodegradable, and noninfectious, and unlike viruses, they do not reproduce in animals, which makes them safe for theragnostic applications [48,51]. Importantly, molecular farming facilitated the large-scale production of plant viruses with high yields and reproducibility that overcome the limitations attributed to producing virus-based delivery systems [48]. Moreover, genetic and protein engineering of plant viruses allows for the fabrication of nanomaterials with a wide range of desirable properties through manipulations of coat protein (CP) subunits.

PVNPs have proven to be highly adaptable and flexible when it comes to integrating and transporting different types of materials. The important parameters for the design of PVNPs are structural properties such as charge, shape, surface addressable groups, and genomes [51]. These properties can be used for loading and targeting cargo via genetic engineering, bioconjugation, infusion, biomineralization, and self-assembly strategies (reviewed in [46]). The internal cavities of these nanoparticles can be used to encapsulate a wide range of cargo using techniques such as self-assembly or infusion. This means that PVNPs are capable of incorporating and delivering various payloads due to the empty spaces within their structure [46,51]. Cargo loading during self-assembly is achieved by changing the buffer conditions. For example, cowpea chlorotic mottle virus (CCMV) disassembles under normal pH or high ionic strength and reassembles under buffer exchange, low pH and low ionic strength to encapsulate CpG oligodeoxynucleotides (ODNs) as an immunotherapy agent [52]. During infusion, cargo loading can be achieved via the spongy property of the genome and surface pores of the capsid. For example, changing conformations under environmental conditions allows the infusion of doxorubicin (DOX) and the lanthanides (Gd3+, Tb3+) into the capsid’s cavity of red clover necrotic mosaic virus (RCNMV) and CPMV, respectively, via changing “open” conformation to “closed” conformation of surface pores and affinity of particles to RNA [53,54]. The process of incorporating mitoxantrone (MTO) into the inner cavity of CPMV (CPMV-MTO) was achieved by means of diffusion through the pores of CPMV [55].

Loading payloads is also possible via the methods of bioconjugation chemistries due to functional addressable groups on interior and exterior surfaces of PVNPs [46,48]. Coupling reactions based on NHS esters, carbodiimides, maleimides, and click chemistries are common protocols for covalent link cargos on these addressable groups. For example, DOX can conjugate via activated carboxylic acid groups on CPMV and Physalis mottle virus (PhMV) [56,57]. The conjugation of immunoglobulin (IgG) isotypes and Trastuzumab to CPMV and the lysine mutant of TMV (TMV-lys) using a strain-promoted azide−alkyne cycloaddition method demonstrated that antibody–VNP conjugates are a stable and functional platform. Trastuzumab-displaying VNPs target HER2-positive SKOV-3 human ovarian cancer cells [58]. A significant development in covalent loading payloads on PVNPs is genetic manipulation. Cargo can be fused to the encoding gene of capsid protein in a single or separate vector expressed by heterogeneous expression systems, simultaneously or separately [46,51].

Non-covalent loading can be achieved via electrostatic interactions between charged cargos and charged addressable groups on the surface of PVNPs. For example, Phenanthriplatin and MTO, a topoisomerase II inhibitor encapsulated by charge-driven drug loading strategies, have been delivered in vivo in a mouse model which demonstrated the superior efficacy of TMV drug vs. free drug [57,59].

## 6. PVNPs as Delivery Nanosystem

Biological barriers reduce the efficacy of therapeutic/diagnostic agents in the body and may lead to severe side effects. PVNP formulations can overcome these limitations by loading into their interior and exterior surfaces and improving their efficacy and safety [46]. Like synthetic nanoparticles, the size, shape, and surface properties affect the therapeutic performances of PVNPs in vivo. For example, the extravasation of elongated PVNPs from the blood vessel is more facile than with spherical PVNPs. Furthermore, it has been observed that PEGylated filamentous PVX is taken up more by human tumor xenografts in comparison to the spherical CPMV [60]. PVNPs can act as a conduit for delivery in passive and active targeting formats (Figure 1).

Structurally, PVNP formulations that are not specifically targeted can achieve the concentration of drug and imaging agents in tumors through passive-targeting delivery by relying on the EPR effect [46,48]. For example, loading cisplatin (cisPt2+) within the interior channel of TMV-based protein nanotubes led to increased cisPt accumulation within tumors versus free cisPt administration, followed by reduced tumor burden, and increased survival of ovarian cancer-bearing mice [62]. Non-targeted PVNPs face a significant obstacle in the form of phagocyte-based clearance, even when PEGylated, due to the presence of anti-PEG antibodies [63]. To counteract this issue, a potential solution is the conjugation of serum albumin (SA) to plant virus-based nanocarriers. In a study conducted on Balb/C mice, it was observed that SA-conjugated TMV, which was “camouflaged” using SA, showed reduced antibody recognition and improved pharmacokinetics [6]. In addition, displaying specific ligands on the surface of PVNPs with high affinity to the target can address these limitations. The ligands of cancer cell-specific biomarkers can conjugate to the PVNP surface via nano-engineering, by targeting PVNP formulations specifically to cancer cells [61]. Several examples of conjugation for targeted delivery of therapeutic agents have been reported in the literature. GE11, a small peptide containing 12 amino acids, has been conjugated to PVX. Similarly, folate has been conjugated to Johnson grass chlorotic stripe mosaic virus (JgCSMV) and pepper mild mottle virus (PMMoV), while Herclon, a monoclonal antibody (mAb) against the inside cell domain of the receptor, HER2, has been conjugated to Sesbania mosaic virus (SeMV)-based VLPs. In addition, PVX-based targeted TNF-related apoptosis-inducing ligand (TRAIL) and peptide F3 on CCMV have been shown to selectively target cancer cells (refer to Figure 2 for more details). The bioconjugation process involving the coupling of epidermal growth factor-like domain 7 (EGFL7), (a protein that is known to be expressed solely in endothelial cells), to CPMV resulted in a modified protein that has the ability to specifically target tumor-associated neovasculature with a high degree of specificity [64]. Recently, a peptide-guided tomato bushy stunt virus (TBSV)-based nanocarrier platform loaded with DOX has been applied for the delivery to specific cells. Marchetti et al. (2023) used TBSV-based VNPs with CooP peptide, a homing peptide to target medulloblastoma tumors. Encapsulating DOX within TBSV-CooP improved cell death and proliferation, demonstrating their efficacy in targeting brain tumors [65]. The internalization of the TBSV-based nanocarrier platform targeted with the C-terminal C-end rule (CendR) peptide, RPARPAR (RPAR) (TBSV-RPAR) loaded with DOX showed selective cytotoxicity towards receptor neuropilin-1 (NRP-1)-expressing cells [66]. Another study demonstrates the use of iRGD peptides to target tumor neovasculature on PhMV-like nanoparticles, resulting in rapid uptake and increased tumor homing. This approach offers a promising platform for targeted molecular cargo delivery to tumors [67]. 

Increasingly, bionanoparticles are being explored for their capability to act as nanocarriers. When compared with synthetic nanoparticles, bionanoparticles are biocompatible and exhibit high target specificity within the host cells, while being non-pathogenic to humans. Antibodies have emerged as important therapeutic molecules and are presently being used for combating several diseases such as autoimmune disorders, cancer, etc. Since antibodies cannot traverse the cellular membrane barrier, intracellular delivery of antibodies is a major challenge.

The coat protein of the icosahedral SeMV was genetically engineered [68] to fuse with the Staphylococcus aureus protein A (SpA) B domain in the region of the βH-βI loop, to synthesize the SeMV loop B (SLB) that self-assembled into VLPs having 43 times greater affinity towards antibodies (~80–90 antibodies/VLP) when compared to SpA alone showing that the chimeric VLPs contained multiple accessible and functional B domains. These VLPs were capable of being internalized into several types of mammalian cells such as B16-F10, BT-474, HeLa, HMECs, and KBs. SLB has demonstrated remarkable proficiency and efficacy in the delivery of three unique mAbs, specifically anti-α-tubulin (designed to target intracellular tubulin), Herclon (designed to counteract the HER2 receptor), and D6F10 (designed to target abrin), directly into the cells. This impressive feat highlights the tremendous potential of SLB to function as a universal nanocarrier for the purpose of intracellular transport of antibodies. Esfandiari et al., 2015, demonstrated the enhancement of antibody cytotoxicity enabled by potato virus X VNPs that were chemically conjugated to Herceptin [69].

## 7. Therapeutic Agent Delivery

Various therapeutic molecules including small molecules (chemotherapy drugs, fluorescent dyes), nucleic acids, peptides, proteins, and nanoparticles can be loaded into PVNPs [46,51]. PVNPs can protect these cargoes from degradation and deliver them, which is conducive to their continued role. PVNPs can load small molecule drugs such as MTO [55], phenanthriplatin [59,70], gemcitabine [71] and cisPt [72], increase their accumulation within the tumor tissue, and induce tumor cytotoxicity. For example, loading DOX by CPMV [56], RCNMV [53], JgCSMV [73], or the prodrug DOX by PhMV-based VLPs [74] significantly improved antitumor efficacy in vitro and in vivo.

TMV disks loaded with DOX were capable of elevating the survival rates of mice harboring intracranial glioblastoma [75]. When cisplatin was loaded onto TMV VNPs by modification of the externally located surface mannose and lactose moieties, this enabled the recognition of the loaded VNPs by the asialoglycoprotein receptor occurring on cell membranes which led to increased cytotoxicity in cancer cell lines [76]. A fluorous molecular ponytail was added at specific sites in the TMV CP [77] that led to the self-assembly of the TMV CP molecules into spherical VNPs. These VNPs, when bound to cisplatin by metal-ligated coordination, showed augmented stability. Also, TMV loaded with phenanthriplatin and cisplatin through charge-based reaction or by means of stable covalent adduct formation could augment absorption by cancer cells which enhanced the cytotoxicity [78]. Valine–citrulline monomethyl auristatin E, an antimitotic drug was loaded onto the external surface of TMV VNPs which resulted in efficacious targeting and improved cytotoxicity in the Karpas 299 non-Hodgkin’s lymphoma cell line in addition to facilitating the entry of the modified VNPs into the endolysosomal compartments followed by the protease-encoded release of the antimitotic drug [79]. Enhanced antitumor effects were observed in murine models upon administration with TMV VNPs loaded with MTO through a charge-driven mechanism [80].

Yin et al., 2012, conjugated the poorly immunogenic tumor-associated Tn carbohydrate antigen (GalNAc-α-O-Ser/Thr) to the Tyr 139 amino acid residue of TMV, which elicited robust immune reactions [81]. In another study, TMV particles were fused with the transacting activation transduction (TAT) peptide on their external surface and these TAT-tagged VNPs were efficiently internalized and enabled RNA silencing in nude mice harboring hepatocellular carcinoma tumors following intratumoral and intravenous delivery [82].

PVNPs can improve poor cell uptake, nuclease-related instabilities, and ineffective delivery limitations of nucleic acids through encapsulating heterologous RNA, siRNAs, mRNA, and CpG-ODNs [61]. For example, brome mosaic virus (BMV) and CCMV can be loaded with the antitumor siRNA Akt1 (siAkt1) for internalization by tumor cells [83]. CCMV, when formulated with siRNAs that target FOXA1, a transcription factor of the forkhead box (FOX) protein family, enables gene silencing in the MCF-7 breast cancer cell line [84]. PVNPs can supply a platform for reducing and improving low stability, and short half-life of amino acid polymer-based therapeutics [61]. They have been used to conjugate and deliver mAb such as Herceptin (Trastuzumab) [85], vascular endothelial growth factor receptor-1 (VEGFR-1) [86], and TRAIL [87].

The use of photothermal therapy (PTT) and photodynamic therapy (PDT) for tumor treatment results in localized shrinkage of the tumor. PVNP-based PTT/PDT agents with adsorption photons can generate heat or reactive oxygen species (ROS) for the ablation of cancer cells [88,89,90]. For example, coating TMV with photothermal biopolymer polydopamine (PDA) and irradiation with near-infrared laser combined with immunotherapy and multimodal magnetic resonance/photoacoustic imaging offers a promising combination and theranostic approach for in vivo cancer models [89,90]. Loading of the porphyrin-based photosensitizer drug, Zn-Por, into TMV and tobacco mild green mosaic virus, TMGMV, has been demonstrated to result in a significant increase in cell-killing efficacy. Specifically, there was a five-fold increase in efficacy observed when compared to the free drug [88].

## 8. Diagnostic Agent Delivery

Nanoengineering of PVNPs offers various opportunities for loading and modifying contrast agents [57]. PVNP-based dyes, known as guanidinium agents are commonly used in preclinical diagnostic imaging. PVNP-based contrast agents have the potential to be developed to accomplish prolonged circulation, specific targeting ability, and effective delivery to tumors in vivo. For instance, one approach involves loading PhMV-like nanoparticles with the fluorescent dye Cy5.5 and paramagnetic Gd(III) complexes, while PEGylated particles can be conjugated with targeting peptides for the purpose of monitoring a human prostate tumor model through near-infrared fluorescence and magnetic resonance imaging [13]. PVNPs can be decorated with bombesin peptides, polyethylene glycol (PEG), and near-infrared fluorescent dyes [91]. By loading Dy^3+^ and Cy7.5 into TMV nanoparticles and conjugating them with a Dy^3+^ dye and near-infrared fluorescence (NIRF) dye, high transverse relaxation of targeted PC-3 prostate cancer cells and tumors was achieved in vitro and in vivo in ultra-high-strength magnetic fields [92].

CPMV loaded with NIR dye (Alexa Fluor 647) and PEG, as well as conjugated with the pan-bombesin analog, [β-Ala11, Phe13, Nle14] bombesin-(7–14) can target the gastrin-releasing peptide receptor that is over-expressed in human prostate cancers. The phenomenon of tumor homing was observed through the utilization of human prostate tumor xenografts on the chicken chorioallantoic membrane model, through intravital imaging techniques [93].

The filamentous PVX can be engineered to display green fluorescent protein (GFP) or mCherry as probes for optical imaging in human cancer cells and within a preclinical mouse model [94]. Plant viruses, specifically TMV, can be the basis for MRI contrast reagents; TMV particles can be loaded with Gd(DOTA) into the interior channel of TMV and the exterior coated with silica, thereby increasing T1 relaxivities compared to uncoated Gd-loaded TMV [95]. The presentation of GE11 on PVX and the conjugation of PVX-GE11 filaments with fluorescent labels can be specifically directed toward the epidermal growth factor receptor (EGFR). The identification and visualization of cells were demonstrated utilizing cell lines of colorectal adenocarcinoma, human skin epidermoid carcinoma, and triple-negative breast cancer (A-431, HT-29, MDA-MB-231), all of which exhibited differing degrees of EGFR upregulation [96].

## 9. Theragnostic Agent Delivery

These unique structural and chemical properties of PVNPs make them highly suitable for combining therapeutic and diagnostic agents’ potentialities for in vivo applications [48]. Metal–phenolic networks (MPNs) based on plant viruses such as TMV, PVX, and CPMV, have been shown to exhibit favorable optical, cytocompatible, and exceptional cell-killing performance in photothermal therapy when loaded with complexes of tannic acid (TA), metal ions (e.g., Fe^3+^, Zr^4+^, or Gd^3+^), or fluorescent dyes (e.g., rhodamine 6G and thiazole orange) and subjected to 808 nm irradiation [97]. Gd-loaded TMV particles coated with the mussel-inspired biopolymer polydopamine (PDA) represent biocompatible nanotheranostic reagents that facilitate multimodal imaging and photothermal therapy (PTT) in PC-3 prostate cancer cells [89]. The capacity of SA-coated tobacco mosaic virus laden with chelated gadolinium (DOTA) for detection via magnetic resonance imaging and the loading of DOX would enable the monitoring of disease progression, thus providing information on the efficacy of the drug delivery strategy [98].

Engineered TMV-MOF (metal–organic framework) hybrid nanoparticles augmented the retention of these VNPs in murine models [99]. Particles of Cy5-TMV@ZIF were produced through the process of coating the TMV that was encapsulated with Cy5 with zeolitic imidazolate framework-8. This resulted in an augmented fluorescence retention time that was 2.5 times higher compared to that of the Cy5-TMV alone. These Cy5-TMV@ZIF particles were resistant to harsh conditions, in addition to being non-toxic and highly stable [99]. Particles of the tobacco mosaic virus (TMV) were impregnated with a metal-free paramagnetic nitroxide organic radical contrast agent (ORCA), resulting in the creation of probes for electron paramagnetic resonance and magnetic resonance imaging for the detection of superoxide. These probes exhibited enhanced in vitro r1 and r2 relaxivities and served as both T1 and T2 contrast agents, thereby illustrating their potential for preclinical and clinical MRI scanning [100].

In yet another study, TMV VNPs were modified to target VCAM-1, the vascular cell adhesion molecule, and these particles were loaded with Gd-dodecane tetraacetic acid (Gd-DOTA). This led to highly sensitive recognition and visualization of atherosclerotic plaques in ApoE-/- mice, employing low doses of contrast agent and this resulted in enhanced relaxivity and moderate tumbling of the Gd-DOTA-TMV carrier with improved signal-to-noise ratio. Moreover, these coupling complexes showed greater imaging sensitivity, thus affording a 40-fold decrease in Gd dose compared to the standard clinical doses [101].

## 10. PVNPs Act as Therapeutic or Adjuvant Agent

Recent studies have shown that some PVNPs alone are immunogenic and highly effective as a monotherapy [49]. Toward this end, several groups have recently demonstrated that intratumoral injection of PVNPs derived from CPMV [14], papaya mosaic virus (PapMV) [102], PVX [103], and alfalfa mosaic virus (AMV) [104] can stimulate antitumor-based immune responses into the tumor microenvironment (TME) as in situ vaccines (ISV). PVNPs overcome the immunosuppressive TME by activating the local innate immune system, restarting the cancer-immunity cycle, and leading to the systemic elimination of cancer cells through the adaptive immune system [12,105]. The antitumor immune stimulation provided by these PVNPs can arise due to their packaged genomes or by their multivalent nature [14,102]. PVNPs as PAMPs are identified by innate immune cells’ pattern recognition receptors (PRRs), including toll-like receptors (TLRs) [49,106]. It has been demonstrated that the antitumor immune stimulation of PapMV depended on RNA [102] and for the empty(e) CMPV on multivalent coat protein assemblies [14]. It has been found that TLR2 and TLR4 are responsible for recognizing eCPMV and TLR7 for RNA-containing CPMV [107]. After binding to surface or endosomal TLRs of antigen-presenting cells (APC), they induce cytokine/chemokine and interferon secretion, to recruit and activate antitumor immune cells [61] (Figure 3). Recently, CPMV with two RNA genomes that have different sizes and unrelated sequences, RNA-1 (6 kb) as the bottom (B) component, and RNA-2 (3.5 kb) as the middle (M) component, can activate innate immune cells to induce the secretion of pro-inflammatory cytokines such as IFNα, IFNγ, IL-6, and IL-12, while inhibiting immunosuppressive cytokines such as TGF-β and IL-10, with the same efficacy as native mixed CPMV [108].

PVNP ISV is a treatment for solid tumors that involves several mechanistic processes. Firstly, the PVNP is transported into the tumor and assimilated by immune cells. Secondly, these activated innate immune cells release cytokines and chemokines to attract more immune cells to combat the tumor. Thirdly, T-lymphocytes become activated and are lured to the tumor to combat tumor cells with their cognate antigens. This results in tumor lysis. Finally, the activated T-lymphocytes travel throughout the body attacking metastatic tumors [105]. For example, the utilization of cowpea mosaic virus (CPMV) as an ISV leads to the upregulation of several immunostimulatory cytokines, namely IL-1β, IL-12, interferon (IFN)-γ, chemokine ligand 3, macrophage inflammatory protein-2, and granulocyte-macrophage colony-stimulating factor [49,109]. Additionally, CPMV-ISV treatment suppresses IL-10 and transforming growth factor β, consequently generating changes in intratumoral cytokines through the altered phenotype of intratumoral myeloid cells. These changes further promote the activation, repolarization, and recruitment of macrophages, dendritic cells (DCs), and neutrophils, all of which exhibit an effector antitumor phenotype. Furthermore, CPMV-ISV treatment significantly enhances effector and memory CD4^+^ and CD8^+^ T cell responses and promotes systemic tumor-specific cytotoxic CD8^+^ T cell activity [110].

In the context of immunotherapy, PVNPs can also act as a nanocarrier for tumor antigens, or immune adjuvants for improving immunotherapeutic efficacy of vaccines [49,105]. In this regard, PVX as an adjuvant can deliver HER2 epitopes to overcome immunological tolerance against HER2 in a PVNP-based peptide vaccine [111]. As an immunomodulator nanocarrier, PVNPs can also overcome unfavorable pharmacokinetic profiles of other immunomodulators. Recently, it has been demonstrated that the incorporation of oligodeoxynucleotides (ODNs, ODN1826) into the CCMV leads to enhanced internalization by macrophages within the tumor microenvironment (TME). This phenomenon results in a deceleration of tumor progression and an extension of survival in murine models of both melanoma and colon cancer [52]. The following Table 1 presents a few reports of PVNP-mediated therapeutic strategies conducted in tumor models. 

## 11. Combined Therapies Based on PVNPs

The integration of PVNP-based monotherapy and any therapy that induces tumor cell lysis and release of tumor-associated antigens (TAAs) might be one of the possible strategies to overcome complexity and tumor heterogeneity [116]. The combination of PVNP monotherapy with radiation therapy (RT) [117], chemotherapy [13,90,103], and immunotherapy [118,119] could form the basis for success. For example, immune checkpoint therapy (ICT) agents and checkpoint-targeting antibodies (e.g., anti-PD-1 antibodies), can break the immunosuppression of T cell activity [120,121]. Combining ICT with CPMV induces an antitumor immune response and increases antitumor efficacy [120]. A combination therapy using CPMV and cyclophosphamide (CPA) has shown remarkable synergistic efficacy against 4T1 mouse tumors in vivo. CPA induced apoptosis and immunogenic cell death (ICD) and synergized with type I interferons elicited by CPMV. Thus, this combination therapy may become a potent new strategy for the treatment of the 4T1 mouse model [13]. It has been suggested that CPMV in combination with RT can turn an immunologically “cold” tumor (with a low number of TILs) into an immunologically “hot” tumor [117]. In PVNP-based combination therapies, PVNP in situ vaccination activates the innate immune system, leading to the recruitment and activation of phagocytes. Other therapies such as chemotherapy, PTT, PDT, and ICT target cancer cells, induce cell death, and boost their capability for cancer cell phagocytosis and in turn, priming the adaptive immune system and leading to potent antitumor immune responses [118]. Loading anticancer peptides (ACPs) into PVNPs provides the capability of disrupting the cell membrane, inducing apoptosis, preventing angiogenesis, and regulating immunity, which can represent a novel strategy for enhancing the effectiveness of cancer therapies [122]. Table 2 shows a list of combination therapies enabled by PVNPs.

## 12. Challenges and Future Perspective of PVNPs

When utilizing PVNPs, there are numerous challenges and limitations to consider. Firstly, in vivo biological barriers such as interactions with serum, immune cells, or antibodies can impact the use of native or functionalized PVNPs in clinical settings. Secondly, anti-PVNP antibodies can alter the way immune cells interact with PVNPs, potentially leading to their elimination before reaching the target site. Thirdly, the formation of protein corona (PC) can impede the development of PVNPs for in vivo applications. Fourthly, since PVNPs often target non-immune cells or tissues, clearance by the immune system should be minimized. Fifthly, expressing certain peptides genetically as part of coat proteins may impair capsid formation, which is necessary for effective infection and PVNP generation in plants. In the context of in situ vaccinations, there is also concern about viral escape from the site of injection, which could result in adverse side effects, in addition to the concern regarding virotherapy and the presence of neutralizing antibodies. Undoubtedly, the presence of existing circulating antibodies can increase the risk of adverse effects.

Although PVNPs are safe for animals, some regulatory issues need to be considered. PVNPs require high throughput manufacturing, dedicated facilities, and adaptations to plant molecular farming. Further research on regulatory pathways, immune system effects, and novel administration methods, such as chimeras or semi-synthetic nanoparticles, is needed. PVNP production processes must adhere to manufacturing standards and quality control measures to ensure safety and consistency, as determined by regulatory agencies.

PVNPs offer significant potential for various therapeutic applications, including but not limited to vaccines, imaging agents, cancer therapies, and drug delivery. The ongoing research endeavors to optimize VNP design, functionalization strategies, and targeting approaches to enhance their therapeutic efficacy. The capacity to modify viruses is constantly expanding, allowing for new therapeutic strategies. By incorporating bioactivatable aspects into the design of innovative viral reagents, researchers can create more flexible and effective treatment options. Directed evolution using mutagenesis strategies can be used to assemble virus libraries that meet specific objectives. Bioinformatics analysis can align different viral capsid genes/protein sequences or protein structures, leading to the generation of novel viruses with unique properties. Computer technologies like machine learning and mathematical modeling can also aid in the development of PVNPs for clinical cancer immunotherapy. In addition, collaborative interdisciplinary partnerships and investments in research infrastructure are crucial in stimulating innovation, expediting scientific progress, and revealing novel possibilities for VNP utilization. Nanoengineering research on VNPs is advancing their design, fabrication, and manufacturing, with various formulations and approaches on the cusp of clinical impact. The structural versatility of VNPs also presents new clinical opportunities across various dose forms. Given the incremental progress of VNP technology in the clinic, investigating pharmaceutical formulation technology may prove instrumental in improving prospects for further translational development. At the preclinical stage, conducting comprehensive research on the formulation of viral nanoparticles (VNP) in various dosage forms would facilitate the modification of VNP’s shelf life stability and compatibility with clinically significant routes of administration. This would in turn enhance the efficacy and safety of the product, while also providing a forecast for future costs and patient convenience.

## 13. Conclusions

Conventional cancer treatment involves chemotherapeutic drug delivery through the circulatory system, causing dispersion and premature drug release before the tumor site. Drug dosage must be significantly increased to ensure efficacy at the target site. This approach may cause unwanted damage to cells and tissues. Precision cancer medicine aims to improve drug efficacy by targeting nanoparticles, demonstrating clinical potential. Plant VNPs are promising due to their self-assembling architecture and easy production. PVNPs offer quality control for uniform particle shape and size, unlike synthetic nanomaterials. Plant VNPs offer biocompatibility, efficacy, low cost, easy manufacturing, and optimal cargo loading capacity. Plant VNPs offer biodegradability, genetic engineering customization, and superior modification efficacy over synthetic nanomaterials, which rely on chemical synthesis and are persistent in the body. The fact that PVNPs can be used as theranostics for cancer detection and treatment could be a game changer for health outcomes in low- and middle-income countries. In these parts of the world, health burdens will increase and strain what little infrastructure is currently available for escalating elderly populations. The potential of plant VNPs could mitigate future health challenges for populations around the planet, particularly in the Global South. It is for this reason that plant VLPs must continue to be under exploration as a plausible solution for future health challenges.

## Figures and Tables

**Figure 1 vaccines-11-01278-f001:**
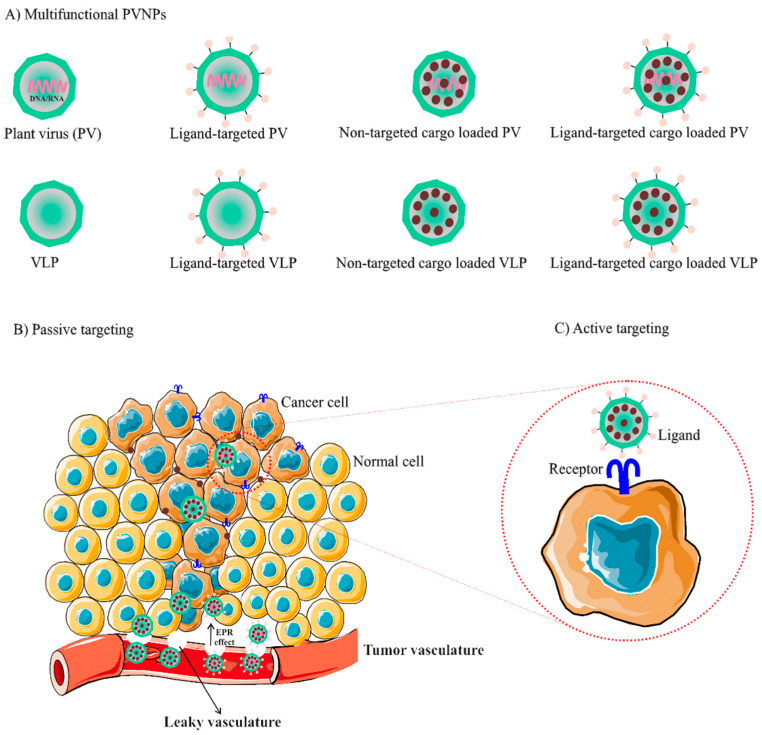
(**A**) The variety of nanoparticle formulations based on plant viruses, (**B**) These formulations of plant virus-based nanoparticles (PVNPs) have the ability to concentrate drug and imaging agents through passive-targeting delivery reliant on the enhanced permeability and retention (EPR) outcome in tumors, (**C**) Active targeting can be achieved by attaching ligands specific to the biomarkers present on cancer cells to the surface of PVNPs, allowing for the specific targeting of cancer cells [61].

**Figure 2 vaccines-11-01278-f002:**
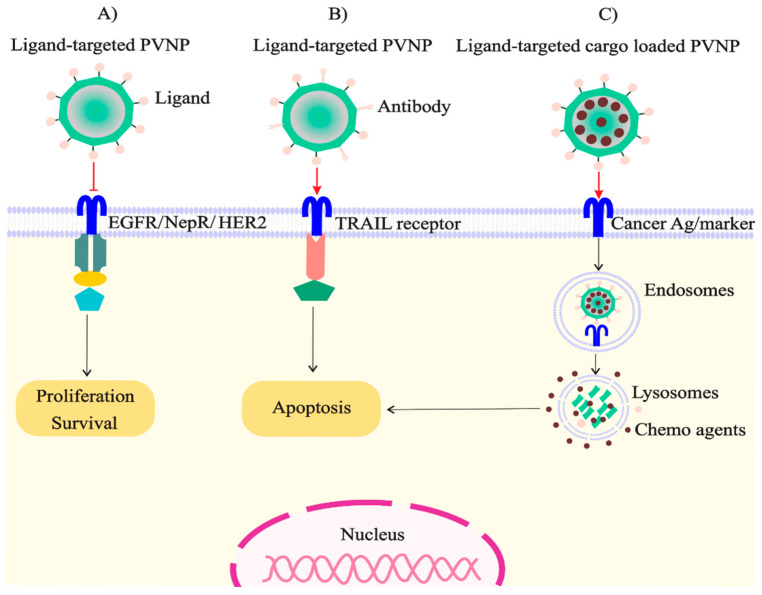
Mechanisms through which plant virus nanoparticles may effectively target cancer cells. The presentation of specific ligands on plant virus nanoparticles has been investigated for the purpose of (**A**) impeding the survival and proliferation pathways, (**B**) instigating the apoptotic pathways, and (**C**) loading and distributing therapeutic agents [61].

**Figure 3 vaccines-11-01278-f003:**
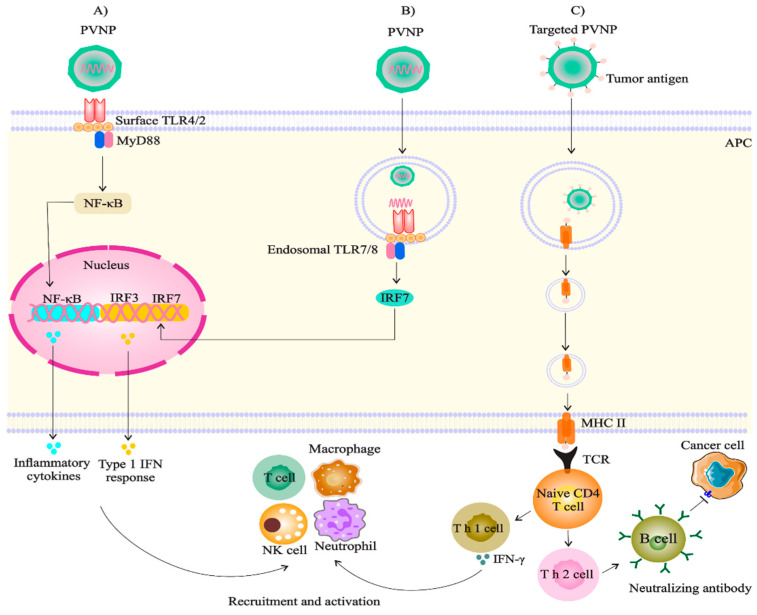
Mechanisms by which plant virus nanoparticles can target immune cells. Plant virus nanoparticles can act as ligands for (**A**) surface and (**B**) endosomal PRRs on innate immune cells, or (**C**) loading and delivery of cancer antigens, therefore active cellular and humoral immune responses against tumor [61].

**Table 1 vaccines-11-01278-t001:** Some examples of therapeutic strategies enabled by PVNPs.

Therapeutic Strategies	PVNP	Delivery Agent	Tumor Modal	Outcomes	Ref.
Monotherapy	CPMV	No agent	B16F10 lung melanoma, ovarian, colon, and breast	Activated innate and adaptive immune systems	[14]
	TMV	No agent	Dermal melanoma	Activated innate and adaptive immune systems	[12]
	AMV	No agent	Breast	Activated innate and adaptive immune systems	[104]
	PapMV	No agent	Melanoma	Activate the innate immune response in an IFN-α-dependent manner	[102]
Vaccine delivery	PVX	Epitopes CH401	Breast	Elevated HER2-specific antibody titers	[112]
	CPMV	Epitopes CH401	Breast	Elevated HER2-specific antibody titers	[112]
	PhMV	CH401	Breast	The delayed onset of tumor growth and the prolonged survival of the vaccinated vs. naïve BALB/C mice	[113]
	CPMV	Ovalbumin	B16F10-OVA	Improved survival and slower tumor growth	[114]
	CPMV	testis antigen NY-ESO-1	NY-ESO-1+ malignancies	CD8^+^ T cells from immunized mice exhibited antigen-specific proliferation and cancer cell cytotoxicity	[115]
Adjuvant delivery	CCMV	CpG	Colon cancer and melanoma	The efficacy of ODN1826 compared to the free drug, slowing tumor growth and prolonging survival	[52]
	TMV	Toll-like receptor 7 agonist (1V209),	B16F10 dermal melanoma i	Greater number of tumor-specific T cells	[90]
	PhMV	CpG-ODN	Breast	Slowing tumor growth and prolonging survival	[113]

**Table 2 vaccines-11-01278-t002:** Examples of PVNP-based combination therapies.

PVNP	Agent	Combination Therapy	Tumor Model	Ref.
CPMV	Anti-PD-1 antibodies, agonistic OX40-specific antibodies, agonistic anti-CD40	Immuno-immunotherapy	Ovarian cancer, colon cancer, and melanoma	[120,123]
CPMV	Irradiated cancer cells (ICCs)	Immuno-immunotherapy	Ovarian cancer	[124]
CPMV	Cyclophosphamide (CPA)	Immuno-chemotherapy	Triple-negative breast cancer (TNBC)	[13]
CPMV	Polydopamine (PDA)	Immuno-photothermal therapy	B16F10 dermal melanoma	[90]
CPMV	Radiation	Immuno-radiation therapy	Ovarian carcinoma	[117]
CPMV	Cryo	Immuno-Cryoablation	Hepatocellular carcinoma (HCC)	[125]
TMV	Porphyrin-based photosensitizer drug (Zn-Por)	Immuno-photodynamic therapy	Melanoma and cervical cancer models	[88]
TMGMV	Porphyrin-based photosensitizer drug (Zn-Por)	Immuno-photodynamic therapy	Melanoma and cervical cancer models	[88]

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
