# Peer review of "Plant Virus Nanoparticles Combat Cancer"

_vaccines, 2023, doi:10.3390/vaccines11081278_

Round 1

Reviewer 1 Report

#The manuscript entitled “Plant Virus Nanoparticles Combat Cancer” is very interesting and can add knowledge for the readers. The utilization of plant virus nanoparticles (VNPs) in cancer immunotherapy has gained significant attention due to their efficacy, safety, biodegradability, and cost-effectiveness. VNPs offer unique advantages in delivering self-antigens and weakly immunogenic antigens, which can help overcome self-antigen tolerance often observed in the tumor microenvironment.

#Additionally, some highly immunogenic VNPs have been shown to stimulate innate immune responses and subsequently initiate adaptive immunity within the tumor microenvironment. The inherent stability and potential for mass production of plant viral VLPs further enhance their appeal as a platform for eliciting anti-tumor immune responses by allowing for antigen and drug modifications.

#Although the overall concept of using plant virus nanoparticles for cancer immunotherapy is promising, there are several serious issues that are needed attention and are essential to consider to make the manuscript publishable. Authors are requested to kindly look at the writing style and also the MDPI style.

#The review should discuss the range of immunogenicity observed with different types of VNPs and their effectiveness in inducing anti-tumor immune responses. It is important to evaluate the preclinical and clinical evidence supporting their efficacy in tumor regression and patient survival.

#While VNPs have demonstrated safety in preclinical models, it is crucial to address potential safety concerns, including immune-related adverse events and unintended immune responses. Additionally, the long-term effects and potential off-target effects of VNPs should be considered and discussed.

# Authors should discuss the strategies employed to achieve tumor-specific targeting of VNPs, as well as their ability to bypass immunosuppressive mechanisms within the tumor microenvironment.

# While plant viral VLPs offer scalability and cost-effectiveness, it is important to address the challenges associated with large-scale production, purification, and quality control to ensure consistent and reproducible VNP formulations for clinical applications.

# Author should explore the potential synergistic effects of combining VNPs with other therapeutic modalities such as immune checkpoint inhibitors, chemotherapy, or radiation therapy. Discuss how these combinations could enhance anti-tumor immune responses and improve overall treatment outcomes.

# Authors should address the current status of clinical trials utilizing VNPs in cancer immunotherapy, highlighting the challenges faced in their translation from preclinical models to human patients. Discuss the regulatory considerations and potential hurdles in bringing VNPs to the clinic.

#By critically examining these aspects, the review can provide a comprehensive assessment of the use of plant virus nanoparticles as innovative tools for cancer immunotherapy, informing researchers and clinicians about their potential and the challenges that need to be addressed for successful clinical application.

#Some additional observations.

The abstract should be rewritten and it should be a summary of the whole manuscript from the aim of this review to the future direction. Every abbreviation should be used only after mentioning the full name first like VLPs.

In the whole manuscript, a few things need to be addressed properly. For example, if any virus name is written in full in the introduction then further only abbreviations should be used like Tobacco mosaic virus (TMV) and Potato virus X (PVX). Here authors somewhere used full names and somewhere used abbreviations, no consistency and no journal pattern

Overall authors are advised to please check the whole manuscript carefully, reorganised the whole content considering the above points, and resubmit.

Moderate editing of the English language required

Author Response

REVIEWER #1 COMMENTS

#The manuscript entitled “Plant Virus Nanoparticles Combat Cancer” is very interesting and can add knowledge for the readers. The utilization of plant virus nanoparticles (VNPs) in cancer immunotherapy has gained significant attention due to their efficacy, safety, biodegradability, and cost-effectiveness. VNPs offer unique advantages in delivering self-antigens and weakly immunogenic antigens, which can help overcome self-antigen tolerance often observed in the tumor microenvironment.

Response: yes, thanks for reviewer suggestion

#Additionally, some highly immunogenic VNPs have been shown to stimulate innate immune responses and subsequently initiate adaptive immunity within the tumor microenvironment. The inherent stability and potential for mass production of plant viral VLPs further enhance their appeal as a platform for eliciting anti-tumor immune responses by allowing for antigen and drug modifications.

Response: yes, thanks for reviewer suggestion.

#Although the overall concept of using plant virus nanoparticles for cancer immunotherapy is promising, there are several serious issues that are needed attention and are essential to consider to make the manuscript publishable. Authors are requested to kindly look at the writing style and also the MDPI style.

Response: yes, thanks for reviewer suggestion.

#The review should discuss the range of immunogenicity observed with different types of VNPs and their effectiveness in inducing anti-tumor immune responses. It is important to evaluate the preclinical and clinical evidence supporting their efficacy in tumor regression and patient survival.

Response: Added 2 tables for immunotherapy

#While VNPs have demonstrated safety in preclinical models, it is crucial to address potential safety concerns, including immune-related adverse events and unintended immune responses. Additionally, the long-term effects and potential off-target effects of VNPs should be considered and discussed.

Response: Added section about challenges of PVNPs for oncotherapy

# Authors should discuss the strategies employed to achieve tumor-specific targeting of VNPs, as well as their ability to bypass immunosuppressive mechanisms within the tumor microenvironment.

Response: Added the following under the section ‘Multifunctional Plant Virus Nanoparticles (PVNPs) in Cancer’.

The important parameter for the design of PVNPs are structural properties (e.g. charge, shape, and surface addressable groups), and genomes {Wen, 2016 #20}. These properties can be use for loading and targeting cargos via genetic engineering, bioconjugation, infusion, biomineralization, and self-assembly strategies (reviewed in {Shahgolzari, 2020 #15}).

Response: Added the following under the section ‘PVNPs act as therapeutic or adjuvant agent’.

PVNP ISV is a treatment for solid tumors that involves several mechanistic processes. Firstly, the PVNP is transported into the tumor and assimilated by immune cells. Secondly, these activated innate immune cells release cytokines and chemokines to attract more immune cells to combat the tumor. Thirdly, T-lymphocytes become activated and are lured to the tumor to combat tumor cells with their cognate antigens. This results in tumor lysis. Finally, the activated T-lymphocytes travel throughout the body attacking metastatic tumors. The utilization of Cowpea mosaic virus (CPMV) as an immune stimulatory virus (ISV) leads to the upregulation of several immunostimulatory cytokines, namely IL-1β, IL-12, interferon (IFN)-γ, chemokine ligand 3, macrophage inflammatory protein-2, and granulocyte-macrophage colony-stimulating factor. Additionally, CPMV-ISV treatment suppresses IL-10 and transforming growth factor β, consequently generating changes in intratumoral cytokines through the altered phenotype of intratumoral myeloid cells. These changes further promote the activation, repolarization, and recruitment of macrophages, dendritic cells (DCs), and neutrophils, all of which exhibit an effector antitumor phenotype. Furthermore, CPMV-ISV treatment significantly enhances effector and memory CD4+ and CD8+ T cell responses and promotes systemic tumor-specific cytotoxic CD8+ T cell activity.

# While plant viral VLPs offer scalability and cost-effectiveness, it is important to address the challenges associated with large-scale production, purification, and quality control to ensure consistent and reproducible VNP formulations for clinical applications.

Response: Added section about challenges of PVNPs for oncotherapy

# Author should explore the potential synergistic effects of combining VNPs with other therapeutic modalities such as immune checkpoint inhibitors, chemotherapy, or radiation therapy. Discuss how these combinations could enhance anti-tumor immune responses and improve overall treatment outcomes.

Response: Added section about combined therapies based on PVNPs

In PVNPs based combination therapies, PVNP in situ vaccination activates the innate immune system, leading to the recruitment and activation of phagocytes. Other therapies such as chemotherapy, PTT, PDT, and ICT targets cancer cells, and induces cell death, and boosts their ability of cancer cell phagocytosis and in turn, priming the adaptive immune system and leading to a potent antitumor immune response.

# Authors should address the current status of clinical trials utilizing VNPs in cancer immunotherapy, highlighting the challenges faced in their translation from preclinical models to human patients. Discuss the regulatory considerations and potential hurdles in bringing VNPs to the clinic.

Response: Added section about challenges, regulatory issues and perspectives of PVNPs for oncotherapy

The capacity to modify viruses is constantly expanding, allowing for new therapeutic strategies. By incorporating bioactivatable aspects into the design of innovative viral reagents, researchers can create more flexible and effective treatment options. Directed evolution using mutagenesis strategies can be used to assemble virus libraries that meet specific objectives. Bioinformatics analysis can align different viral capsid genes/protein sequences or protein structures, leading to the generation of novel viruses with unique properties. Computer technologies like machine learning and mathematical modeling can also aid in the development of PVNPs for clinical cancer immunotherapy.

#By critically examining these aspects, the review can provide a comprehensive assessment of the use of plant virus nanoparticles as innovative tools for cancer immunotherapy, informing researchers and clinicians about their potential and the challenges that need to be addressed for successful clinical application.

Response: Added section about challenges and future perspectives of PVNPs for oncotherapy

#Some additional observations.

The abstract should be rewritten and it should be a summary of the whole manuscript from the aim of this review to the future direction. Every abbreviation should be used only after mentioning the full name first like VLPs.

Response: Abstract re-written as per your kind suggestion.

In the whole manuscript, a few things need to be addressed properly. For example, if any virus name is written in full in the introduction then further only abbreviations should be used like Tobacco mosaic virus (TMV) and Potato virus X (PVX). Here authors somewhere used full names and somewhere used abbreviations, no consistency and no journal pattern

Response: Above concern regarding abbreviations addressed.

Overall authors are advised to please check the whole manuscript carefully, reorganised the whole content considering the above points, and resubmit.

Response: Manuscript edited as per your kind advice.

Reviewer 2 Report

Nicely constructed and well-written on novel topic. Please mention in a section how, these nanoparticles will be clinically translated from lab to hospitals in near future. Also write an section on could this plant virus nanoparticles could be combined with nanodrug delivery of conventional chemotherapeutics agents to treat cancers as well. Give some of the novel perspective. 

Author Response

Nicely constructed and well-written on novel topic. Please mention in a section how, these nanoparticles will be clinically translated from lab to hospitals in near future. Also write a section on could this plant virus nanoparticles could be combined with nanodrug delivery of conventional chemotherapeutics agents to treat cancers as well. Give some of the novel perspective. 

Response: Added one paragraph to the section on the ‘Challenges, regulatory issues and future perspective of PVNPs’ that covers clinical aspects. Also, table 2 offers some combination of PVNPs with other therapies that were also suggested by Reviewer #1.

Reviewer 3 Report

Overall, this review highlights the remarkable versatility and potential of plant viruses, specifically Plant virus nanoparticles, in addressing various health challenges. The review emphasizes the need to further explore the possibilities of plant VNPs and acknowledges their potential to mitigate future health challenges, making a strong case for their continued investigation.

The review described about the VNPs importance in Therapeutic agent delivery, Diagnostic agent delivery, Theragnostic agent delivery, and PVNPs act as therapeutic or adjuvant agent.

Overall the review article could be accepted in present format

Author Response

No comments to be addressed. 

Round 2

Reviewer 1 Report

Authors have thoroughly revised manuscript and now it can be accepted

Author Response

Thank you very much for your kind consideration.